# Advanced molecular surveillance approaches for characterization of blood borne hepatitis viruses

Michael G. Berg[1]*, Ana Olivo[1], Kenn Forberg[1], Barbara J. Harris[1], Julie Yamaguchi[1], Rachel Shirazi[2], Yael Gozlan[2], Silvia Sauleda[3,4], Lazare Kaptue[5], Mary A. Rodgers[1], Orna Mor[2,6]°, Gavin A. Cloherty[1]°

1 Infectious Diseases Research, Abbott Diagnostics, Abbott Park, Illinois, United States of America,
2 Central Virology Laboratory, National HIV and Viral Hepatitis Reference Center, Public Health Services, Ministry of Health, Tel-Hashomer, Ramat-Gan, Israel, 3 Transfusion Safety Laboratory, Banc de Sang i Teixits, Servei Català de la Salut, Barcelona, Spain, 4 Centro de Investigación Biomédica en Red de Enfermedades Hepáticas y Digestivas (CIBERehd), Instituto de Salud Carlos III, Madrid, Spain, 5 Université des Montagnes, Bangangté, Cameroon, 6 Sackler Faculty of Medicine, Tel Aviv University, Tel-Hashomer, Israel

° These authors contributed equally to this work.
* michael.berg@abbott.com

**Data Availability Statement:** Sequence data has been deposited in Genbank. Accession numbers for HCV sequences are MT632105-MT632194. Accession numbers for HBV sequences are

## Abstract

Defining genetic diversity of viral infections directly from patient specimens is the ultimate goal of surveillance. Simple tools that can provide full-length sequence information on blood borne viral hepatitis viruses: hepatitis C, hepatitis B and hepatitis D viruses (HCV, HBV and HDV) remain elusive. Here, an unbiased metagenomic next generation sequencing approach (mNGS) was used for molecular characterization of HCV infections (n = 99) from Israel which yielded full-length HCV sequences in 89% of samples, with 7 partial sequences sufficient for classification. HCV genotypes were primarily 1b (68%) and 1a (19%), with minor representation of genotypes 2c (1%) and 3a (8%). HBV/HDV coinfections were characterized by suppressed HBV viral loads, resulting in sparse mNGS coverage. A probe-based enrichment approach (xGen) aiming to increase HBV and HDV coverage was validated on a panel of diverse genotypes, geography and titers. The method extended HBV genome coverage a median 61% (range 8–84%) and provided orders of magnitude boosts in reads and sequence depth for both viruses. When HBV-xGen was applied to Israeli samples, coverage was improved by 28–73% in 4 samples and identified HBV genotype A1, A2, D1 specimens and a dual B/D infection. Abundant HDV reads in mNGS libraries yielded 18/26 (69%) full genomes and 8 partial sequences, with HDV-xGen only providing minimal extension (3–11%) of what were all genotype 1 genomes. Advanced molecular approaches coupled to virus-specific capture probes promise to enhance surveillance of viral infections and aid in monitoring the spread of local subtypes.

MT622522-MT622525. Accession numbers for
HDV sequences are MT583788-MT583813.

**Funding:** The authors declare we were funded by
Abbott Laboratories and that one or more of the
authors have an affiliation to the commercial
funders of this research study, Abbott
Laboratories. The funder provided support in the
form of salaries for authors [MGB, AO, KF, BJH,
JY, MAR, GAC], but did not have any additional
role in the study design, data collection and
analysis, decision to publish, or preparation of the
manuscript. The specific roles of these authors are
articulated in the 'author contributions' section.

**Competing interests:** MGB, AO, KF, BJH, JY, MAR,
GAC are all employees and shareholders of Abbott
Laboratories. This commercial affiliation does not
alter our adherence to all PLOS ONE policies on
sharing data and materials. Sequences generated
in this study have been deposited in GenBank. No
patents have been applied for and no products are
in development related to this research.

## Introduction

Viral hepatitis represents a significant global health burden, particularly as many cases lead
to cirrhosis and liver cancer, which can be fatal. Viral surveillance is essential to understand
prevalence and determine appropriate public health measures. In particular, hepatitis B
virus (HBV), hepatitis C virus (HCV), and hepatitis delta virus (HDV) are major burdens on
human health worldwide that must be monitored. Given the large numbers of genotypes and
sub-genotypes for all three of these hepatitis viruses, the wide spectrum of genetic diversity
they encompass brings the inherent potential to evade detection by diagnostic tests [1–3].
While generation of partial sequences for a given viral genome by Sanger sequencing methods
has provided the classical means of surveillance, there are several drawbacks to consider
with this approach. First, sub-genomic sequences can underestimate true diversity in a popu-
lation that may contain recombinant strains. Second, focused sequencing in one region may
not adequately inform diagnostic assay development that targets un-sequenced regions of
the virus. Third, this method requires the design of primers for amplification that may not
work for all genotypes. Alternatively, these issues can be avoided by pursuing complete
genome sequencing.

The application of next-generation sequencing (NGS) to obtain full genomes is an invalu-
able epidemiological tool for tracking where strains have traveled, identifying transmission
networks, spotting an outbreak, and monitoring for mismatches in diagnostics [4–8]. Unbi-
ased metagenomics using random priming permits any pathogen to be detected in patient
specimens, including viruses, bacteria, parasites, and fungi [9, 10]. However, abundant host
background reads can obscure the presence of many of these agents and these methods are
challenged by small, diverse, low copy, and or highly structured hepatitis viruses, such as HBV
and HDV. Target enrichment offers an opportunity to significantly boost sensitivity, resulting
in improved coverage and higher confidence data [11]. Single stranded DNA probes can be
hybridized to reads within mNGS libraries to selectively capture and amplify viral sequences
and is particularly useful for samples with low viral loads [12, 13]. A post-library capture step
(e.g. xGen) has successfully been deployed for blood borne RNA viruses in the 10 kb-range
length, like HIV and HCV, although it has not yet been evaluated for small viruses like HBV
(3.2 kb) and HDV (1.6 kb) [13, 14].

Molecular surveillance of hepatitis viruses is particularly important in Israel, where immi-
gration and travel rates are high. The countrywide prevalence of chronic HBV (HBsAg+) is
estimated at 1.75%, with HDV co-infection at 6.5–7.1% [15, 16]. In general, the eastern Medi-
terranean region has the highest levels of HCV at nearly 2.5%, but large studies specific to
Israel puts country prevalence at 0.5–0.9%, largely due to eastern European and Russian immi-
grants from the former Soviet Union [16–19]. While the incidence of newly-diagnosed cases
has been declining and most infections are Genotype 1b, Israel is home to numerous immi-
grant populations with the capacity to import new strains of HCV [19, 20]. To date, circulating
strains for all three viruses have largely been determined by sub-genomic sequencing [15, 21].
Therefore, we applied metagenomic and two target enrichment NGS techniques (xGen and
Pan viral probes) to study epidemiological trends of HCV-infected and HBV/HDV co-infected
individuals in Israel.

## Materials and methods

### Specimens

Patient plasma was collected from individuals seeking treatment at the Israeli National HIV
and Viral Hepatitis Reference Center (NHRL). Plasma samples were remains from patients

referred to the laboratory for HDV viral load measurements (HBV-HDV co-infections) or for HCV RAS analysis. HDV positivity was defined by real-time PCR [15]. All specimens were de-identified and IRB approval was granted for mNGS. Patients were exempt from signing a consent form by the local IRB (approval numbers for HCV and for HDV are 9329-12-SMC and 2890-15-SMC, respectively). Specimens in the HBV genotype panel were purchased from Boca Biolistics (Pompano Beach, FL) or collected from volunteer blood donors in Spain and Cameroon (HBV/HDV co-infected) to demonstrate probe efficacy on diverse strains. Spanish samples were selected from the Biobank of the Catalonia Blood Bank and de-identified. Participants were recruited from Barcelona and provided written, informed consent. IRB approval was obtained from the Vall d'Hebron Hospital Ethics Committee. Cameroonian samples were from two HBV surveillance studies conducted from 2010–2016 where participants were recruited from blood bank donors, hospitals, and chest clinics in the urban centers of Douala and Yaoundé. Written informed consent was provided and plasma was collected anonymously. Studies were approved by the Ministry of Health of Cameroon, the Cameroon National Ethical Review Board, and the Faculty of Medicine and Biomedical Science IRB. Israeli HDV samples were collected from October 2014-Jul 2017 and HCV samples were collected from June 2015-Jul 2017. Only DAA naïve HCV-RNA positive patients and HBV+ HDV patients with detectable HDV RNA were recruited into the study. Samples were randomly collected from these patients and can be considered representative of a larger HCV positive or HBV/HDV dually positive population. Relevant demographic details are included in S1 Table. Negative controls were normal human plasma (NHP) and positive controls consisted of purified stocks or infected plasma in which virus or bacteria was diluted into NHP to log 4.0 copies/ml. Parvovirus B19, HHV-5, VZV, Influenza A, Adenovirus 7, and Chlamydia stocks were from Exact Diagnostics (Dallas, TX); HIV, HBV, and HCV originated from samples sourced in Cameroon and Spain.

## Viral loads

HDV and HBV viral loads (22 samples) were determined by quantitative PCR in Israel [15]. HCV and HBV viral loads were approximated at Abbott using a semi-quantitative multiplex PCR. This research-use assay simultaneously detects HBV, HCV, HIV-1, and HIV-2. Quantitation was extrapolated from relative Ct values of diluted standards.

## Specimen pretreatment and extraction

Plasma specimens were pre-treated with Ultra-pure benzonase (Sigma, St. Louis MO) for 3 hrs at 37˚C and extracted on an $m2000sp$ (Abbott Laboratories, Des Plaines IL) using the RNA/DNA protocol (500 μl input/50 μl elute).

## mNGS library production

Metagenomic libraries (mNGS) were prepared and quantified essentially as described [13]. Briefly, total nucleic acid was concentrated to 10 μl with RNA Clean and Concentrator-5 spin columns (Zymo Research, CA) and RNA was reverse transcribed with random primers using Superscript III (SSRTIII) 1st Strand reagents (Life Technologies), followed by 2nd strand synthesis with Sequenase V2.0 T7 DNA pol (Affymetrix). Double stranded DNA/cDNA was recovered with DNA Clean and Concentrator-5 spin columns (Zymo Research) and -barcoded with Nextera XT indices lacking 5' biotin tags using 24 cycles of amplification (IDT, Coralville IA; Illumina, Carlsbad CA). Nextera libraries were purified with Agencourt AMPpure XP beads (Beckman Coulter) and quantified by a 2200 TapeStation (Agilent) and Qubit fluorometer (Life Technologies).

## Design of HBV and HDV xGen probe sets

Probe sets were designed essentially as described previously for HIV, with each probe 120 nt in length [13]. Briefly, 60 HBV complete genomes including genotypes A-I were aligned in BioEdit. A single consensus (3223 nt) was extracted, with degenerate bases replaced by specific nucleotides, and an initial 53 probes at 2X coverage (e.g. 60 nt overlap) were designed from this sequence. The alignment was surveyed in 120 nt windows to identify regions with <80% identity and include any genotype-specific fragments each 120, 239, or 257 nt in length. An additional 25 probes with 1X coverage (e.g. 1 nt overlap) were designed for a total of 78 HBV probes. For HDV, sequence diversity is much greater, requiring probes designed from separate consensus genomes of genotypes 1–8. Genomes ranged from 1293 nt-1693 nt in length, resulting in 109 probes at approximately 1X coverage. An additional 28 HDV probes in diverse regions were included for a total of 137. HBV and HDV probe sets were combined into one reagent (215 probes) for hybridization.

## xGen reagent synthesis and protocol

120 nt probe stocks (3 pmol/probe) modified with a 5' biotin tag, Nextera barcoding primers lacking a biotin label, and blocking oligos complementary to Nextera Set A i5 and i7 index primers (1 μl/rxn) were all synthesized at IDT. Hybridizations, capture by streptavidin beads, washes and library amplification were essentially as described [13]. Here, after the initial 12 cycles of amplification and elution off streptavidin beads, a repeat KAPA amplification of 10 cycles was performed and libraries were visualized on a 2200 TapeStation and quantified with a Qubit fluorometer using the dsDNA high-sensitivity kit.

## Pan-viral enrichment

SSRTIII-Nextera (mNGS) libraries from the HBV diversity panel were pooled together for enrichment with the commercially available Pan Viral probe set (n>600,000 probes). Hybridization and amplification steps were followed according to manufacturer instructions (Twist Biosciences, San Francisco, CA). A complete description of the procedure is included in the S1 Appendix. We note that after DNA purification of library amplification on streptavidin beads, a second 15 cycle PCR 'off the beads' was performed.

## Next generation sequencing and analysis

HCV dual barcoded libraries were multiplexed according to viral load and either sequenced on a single HiSeq run (n = 72) or batched into 4 runs of 7 (n = 28) on a MiSeq to achieve sufficient read depth. HBV and HDV mNGS libraries (n = 26) were divided over 4 MiSeq runs. xGen libraries were pooled together for a single, separate run, since these share the same barcodes as mNGS libraries. NGS data analysis was performed as described with CLC Genomics Workbench 9.0 software (CLC bio/Qiagen, Aarhus Denmark) and SURPI [22, 23]. Raw data was initially mapped to multiple reference sequences to determine the genotype with the greatest identity. An iterative approach was used to derive the final sequence, using the initial consensus as the reference to refine the consensus upon remapping. To detect possible contaminating reads from barcode hopping, raw data from each sample was individually mapped to the consensus sequences of other samples sequenced on the same run, removing any with ≥99% identity. Unmapped reads (e.g. unique to the sample of interest) were collected and realigned to generate the final consensus.

## Phylogenetic analysis

Multiple sequence alignments were performed in MegAlign Pro (Lasergene, DNASTAR Inc., Madison WI) using the MUSCLE algorithm and manually edited in BioEdit Sequence Alignment Editor (v 7.2.5) [24, 25]. Neighbor-Joining phylogenetic inference was performed using PHYLIP (version 3.5c; J. Felsenstein, University of Washington, Seattle, WA). Evolutionary distances were estimated with Dnadist (Kimura two-parameter method) and phylogenetic relationships were determined by Neighbor (neighbor-joining method). Branch reproducibility of trees was evaluated using Seqboot (100 replicates) and Consense. Trees were visualized using FigTree (version 1.4.2; A. Rambaut, Institute of Evolutionary Biology, University of Edinburgh, Edinburgh). Sequences basal to a genotype branch were examined for recombination breakpoints with SimPlot (version 3.5.1; S. Ray, Johns Hopkins University, Baltimore, MD).

## Nucleotide sequence accession numbers

Full-length sequences were submitted to GenBank under the following accessions: HCV (MT632105-MT632194), HBV (MT622522-MT622525), and HDV (MT583788-MT583813).

# Results

## Full genome coverage of HCV without enrichment

A total of 99 HCV RNA positive specimens were collected at the NHRL. A semi-quantitative PCR of libraries indicated that the average viral load was log 5.4 copies/ml, consistent with previously observed trend of most untreated HCV patients having viral loads >4 log IU/ml [26]. A median of 8.3 million reads were obtained per library (S2 Table). Full length HCV sequences ($\geq$97% coverage) were obtained for 82 individuals and $\geq$90% of the genome was determined for 89/99 samples (Table 1). Five of these had 62–86% coverage despite viral loads $\geq$ log 5, whereas the remaining 5 with < 50% coverage had either a viral load < log 5 or produced very few total mNGS reads. Numbers of reads and the level of coverage depth provide an indication of the confidence in the consensus sequences. HCV reads were normalized to total reads and expressed as reads/million (rpm). The majority (93%) of mNGS libraries had $\geq$100 HCV rpm, with most (63%) having $\geq$1000 HCV rpm (Table 1). A median of 11,466 HCV reads were obtained per library. Due to the high coverage obtained for HCV specimens, no further enrichment was necessary.

Full-length consensus sequences were added to a complete alignment of HCV references and analyzed within neighbor joining phylogenetic trees (Fig 1). HCV genotypes were primarily 1b (68%) and 1a (19%), with minor representation of genotypes 2c (1%) and 3a (8%) (Table 1). Branch lengths indicated all Israeli strains (red) were unique; nevertheless, those

**Table 1. HCV mNGS metrics.**

| % Genome Coverage | Total individuals | HCV Reads/Million | Total individuals | HCV Genotype | Total individuals |
|---|---|---|---|---|---|
| $\geq$97% | 82 | 10,000–100,000 | 12 | 1a | 19 |
| 90–97% | 7 | 1,000–10,000 | 51 | 1b | 68 |
| 50–90% | 5 | 100–1,000 | 30 | 2c | 1 |
| 20–50% | 2 | 10–100 | 4 | 3a | 8 |
| <20% | 3 | <10 | 2 | n.d. | 3 |

The first two columns report the breakdown of samples according to genome coverage. The second two columns report the breakdown of samples by HCV reads per million. The third two columns report the breakdown of samples by genotype.

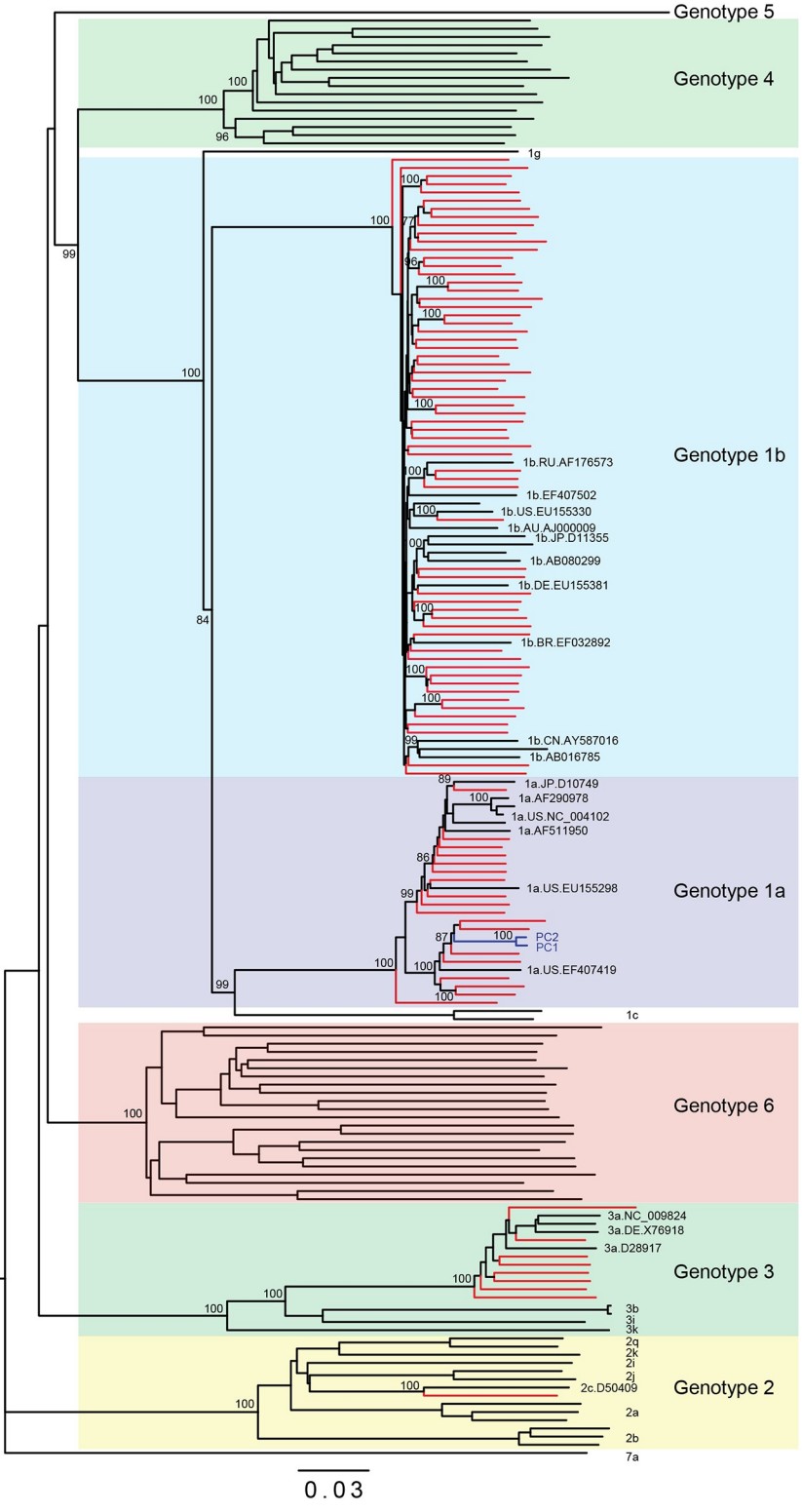

**Fig 1. The majority of Israeli strains are Genotype 1b.** Neighbor-joining phylogenetic tree of full-length HCV sequences. New Israeli sequence branches are in red and reference strains are in black labeled with accession numbers. Bootstrap values of nodes >70 are shown.

branching with high bootstrap values were aligned pairwise to rule out cross-contamination. These sequences shared only 91–94% identity and none shared a barcode. The basal branching pattern of Genotype 1b strain (sample 25–2000618) was evaluated in Simplot but did not show evidence of recombination. Genotype 3a sequences had highest identity (≤95%) with strains originating in Western Europe. These strains generally came from male, injection drug users with a median age of 49 ± 7 years, and not from a specific country. For the 7 partial genomes (20–90% coverage), separate trees were generated from gap-stripped alignments and there was still sufficient sequence to allow for classification.

mNGS output for all 99 original samples were analyzed with the SURPI metagenomics pipeline to probe for additional viruses, such as HPgV-2, a recently appreciated co-infection of HCV [23, 27]. A variety of viruses (e.g. HIV, HBV, VZV, Influenza, adenovirus, etc.) spiked into normal human plasma each at log 4.0 copies/ml served as a positive control and all were detected. Coverages and reads per million values were similar for the three positive controls included in the separate sequencing runs of HCV libraries (S1 Fig). Human pegivirus-1 (GBV-C) reads were enriched in 5 individuals. No additional blood borne agents or viruses besides HCV were found in any of the 99 mNGS patient libraries.

## Target enrichment of HBV and HDV

HDV antibody positive HBV/HDV specimens with detectable HDV RNA by quantitative PCR were prepared in the same manner as HCV for mNGS. Preliminary viral loads measured independently in Israel and at Abbott indicated that levels of co-infecting HBV were very low, with only a minority of the 26 samples registering <33 Ct (1–3 log copies/ml) and most without detectable levels of DNA [15]. Consequently, HBV coverage by mNGS was sparse, if not completely absent (Table 2). HDV by contrast had higher viral loads (4–7 log copies/ml) and was readily detected in 100% of samples, with coverages ranging from 20–100%.

To improve target identification, 5' biotin-tagged xGen probes each 120 nt in length were designed to include the entire spectrum of HBV (genotypes A-I) and HDV (genotypes 1–8) genetic diversity. Our present samples only required enrichment for HBV, but given that probes to other strains or viruses do not interfere with each other, we combined these into one probe set (n = 215) for future use (Fig 2A). The HBV-HDV xGen probe methodology was first validated on 12 unrelated mono- and co-infected samples from a variety of countries that are known to include a diverse array of genotypes. Since xGen libraries derive from mNGS libraries and thus have the same barcodes, they were sequenced on different runs to be able to discriminate the source of reads. Coverage plots of an HBV genotype F1 (1007-HBV-0036) from Peru with a viral load of 3.6 log copies/ml demonstrate the dramatic improvement with probe enrichment (Fig 2B, left). Only 6% coverage at 1X depth was obtained by mNGS, which increased to 90% at 27X depth with xGen. A modest improvement was observed for an HDV genotype 1 with a 4.59 log cp/ml viral load (U160953A), increasing from 90% to 94% coverage, but primarily with an increase in depth (Fig 2B, right). Note that in this sample, HBV (genotype B2) present at 3.24 log IU/ml increased from 8% coverage with mNGS to 84% with xGen (Fig 2C, Table 3).

The boost in sensitivity over a range of viral loads and genetic diversity is summarized for each virus by comparing genome coverage (top) and reads per million (bottom) ± xGen enrichment (Fig 2C, Table 3). While samples with very low viral loads did see some increase for both metrics, those with titers ≥3.5 log saw the most improvement. The median increase in HBV genome coverage was 61% (range 8–84%). The same Nextera mNGS libraries obtained from mono- and co-infected samples from a variety of countries were similarly captured and amplified with a "Pan-viral" probe set which contains >600,000 probes against 1000 human

**Table 2. Israeli HBV/HDV mNGS and xGen metrics.**

| Specimen ID | Virus | Genotype | Viral load / Ct | Final sequence length (nt) | Bootstrap | Library Type | Total Reads | HBV or HDV reads | % virus reads | Genome Coverage% | Avg coverage depth |
|---|---|---|---|---|---|---|---|---|---|---|---|
| 2000166 | HBV | B2 | - | 203 | 89 | mNGS | 526,669 | 211 | 0.04% | 6% | 41 |
| - | - | - | *not detected* | - | - | xGen | 8,105 | 2 | 0.02% | 2% | 1 |
| - | HDV | 1 | - | 654 | 100 | mNGS | 599,442 | 10 | 0.002% | 48% | 1.48 |
| - | - | - | - | - | - | xGen | 9,056 | 1,594 | 17.60% | 40% | 138.03 |
| 2000208 | HBV | nd | 148.45 | nd | nd | mNGS | 1,221,334 | 0 | 0% | 0% | 0 |
| - | - | - | *34.64* | - | - | xGen | 154,910 | 0 | 0% | 0% | 0 |
| - | HDV | 1 | 1150000 | 1541 | 100 | mNGS | 1,221,334 | 6,859 | 0.56% | 94% | 406.72 |
| - | - | - | - | - | - | xGen | 154,910 | 137,271 | 88.61% | 96% | 8390.27 |
| 2000234 | HBV | nd | - | nd | nd | mNGS | 354,688 | 20 | 0.01% | 1% | 0.17 |
| - | - | - | - | - | - | xGen | 5,856 | 2 | 0.03% | 1% | 1.68 |
| - | HDV | 1 | 248000 | 894 | 100 | mNGS | 354,688 | 189 | 0.05% | 50% | 21.37 |
| - | - | - | - | - | - | xGen | 5,856 | 1,124 | 19.19% | 62% | 95.74 |
| 2000236 | HBV | nd | 5.09 | 185 | nd | mNGS | 2,006,742 | 434 | 0.02% | 2% | 104 |
| - | - | - | *37.11* | - | - | xGen | 970,261 | 226,990 | 23.39% | 4% | 71192 |
| - | HDV | 1 | 2140000 | 1600 | 100 | mNGS | 2,328,818 | 12,717 | 0.55% | 94% | 896.85 |
| - | - | - | - | - | - | xGen | 1,006,436 | 440,930 | 43.81% | 96% | 32627.18 |
| 2000319 | HBV | nd | 0.98 | nd | nd | mNGS | 827,658 | 0 | 0% | 0% | 0 |
| - | - | - | *42.94* | - | - | xGen | 11,286 | 0 | 0% | 0% | 0 |
| - | HDV | 1 | 54750 | 1419 | 100 | mNGS | 827,658 | 43 | 0.01% | 79% | 4.1 |
| - | - | - | - | - | - | xGen | 11,286 | 77 | 0.68% | 79% | 6.78 |
| 2000320 | HBV | nd | Indeter. | 58 | nd | mNGS | 61,370 | 2 | 0.003% | 1% | 1 |
| - | - | - | - | - | - | xGen | 5,556 | 0 | 0% | 0% | 0 |
| - | HDV | 1 | 173563 | 1119 | 100 | mNGS | 65,364 | 5 | 0.01% | 20% | 2.18 |
| - | - | - | - | - | - | xGen | 5,556 | 4,342 | 78.15% | 30% | 1240.82 |
| 2000322 | HBV | nd | 410.94 | 138 | nd | mNGS | 1,505,468 | 1 | 0.0001% | 2% | 1 |
| - | - | - | *33.08* | - | - | xGen | 2,349,944 | 1 | 0.0000% | 2% | 0.01 |
| - | HDV | 1 | 40000 | 1641 | 100 | mNGS | 1,505,468 | 6,048 | 0.40% | 95% | 369.02 |
| - | - | - | - | - | - | xGen | 2,349,944 | 1,144,523 | 48.70% | 98% | 84759.81 |
| 2000324 | HBV | nd | - | 85 | nd | mNGS | 1,068,387 | 120 | 0.01% | 2% | 62 |
| - | - | - | *not detected* | - | - | xGen | 312,189 | 7,363 | 2.36% | 2% | 3561 |
| - | HDV | 1 | - | 1210 | 100 | mNGS | 1,154,948 | 467 | 0.04% | 87% | 32.5 |
| - | - | - | - | - | - | xGen | 313,010 | 242,129 | 77.36% | 85% | 15973.34 |
| 2000372 | HBV | nd | - | 139 | nd | mNGS | 1,933,208 | 380 | 0.02% | 2% | 26 |
| - | - | - | *not detected* | - | - | xGen | 2,167,747 | 26 | 0.001% | 4% | 6 |
| - | HDV | 1 | 21800000 | 1642 | 100 | mNGS | 2,215,648 | 62,193 | 2.81% | 90% | 4319.08 |
| - | - | - | - | - | - | xGen | 2,236,978 | 1,060,712 | 47.42% | 98% | 72427.13 |
| 2000563 | HBV | nd | 16.58 | 66 | nd | mNGS | 522,433 | 111 | 0.02% | 2% | 25 |
| - | - | - | *35.57* | - | - | xGen | 79,266 | 0 | 0% | 0% | 0 |
| - | HDV | 1 | 1.4x10(8) | 1471 | 100 | mNGS | 580,988 | 1,747 | 0.30% | 90% | 133.92 |
| - | - | - | - | - | - | xGen | 79,266 | 61,293 | 77.33% | 90% | 4422.76 |
| 2000570 | HBV | D, basal | Indeter. | 449 | 75 | mNGS | 120,604 | 0 | 0% | 0% | 0 |
| - | - | - | - | - | - | xGen | 165,848 | 904 | 0.55% | 14% | 96 |
| - | HDV | 1 | 1.24x10(9) | 1676 | 100 | mNGS | 120,604 | 11,661 | 9.67% | 90% | 1039.97 |
| - | - | - | - | - | - | xGen | 165,436 | 96,813 | 58.52% | 98% | 7365.02 |
| 2000741 | HBV | nd | - | nd | nd | mNGS | 376,532 | 0 | 0% | 0% | 0 |
| - | - | - | *41.06* | - | - | xGen | 2,400 | 0 | 0% | 0% | 0 |

*(Continued)*

**Table 2.** (Continued)

| Specimen ID | Virus | Genotype | Viral load / *Ct* | Final sequence length (nt) | Bootstrap | Library Type | Total Reads | HBV or HDV reads | % virus reads | Genome Coverage% | Avg coverage depth |
|---|---|---|---|---|---|---|---|---|---|---|---|
| - | HDV | 1 | - | 1103 | 100 | mNGS | 376,532 | 32 | 0.01% | 55% | 4.1 |
| - | - | - | - | - | - | xGen | 2,400 | 15 | 0.63% | 40% | 2.24 |
| 2000742 | HBV | A1 | 218.75 | 1810 | 79 | mNGS | 2,171,531 | 38 | 0.002% | 13% | 1 |
| - | - | - | *35.07* | - | - | xGen | 33,656 | 1,361 | 4.04% | 54% | 26 |
| - | HDV | 1 | 44800 | 1510 | 100 | mNGS | 4,415,132 | 1,180 | 0.03% | 93% | 43.65 |
| - | - | - | - | - | - | xGen | 37,374 | 28,799 | 77.06% | 90% | 1380.97 |
| 2000744 | HBV | D | 984.64 | 1779 | 100 | mNGS | 2,342,415 | 179 | 0.01% | 5% | 1 |
| - | - | - | *32.80* | - | - | xGen | 4,680,211 | 1,697 | 0.04% | 55% | 58 |
| - | HDV | 1 | 87400 | 1634 | 100 | mNGS | 2,835,334 | 17,587 | 0.62% | 94% | 1286.69 |
| - | - | - | - | - | - | xGen | 4,767,854 | 1,734,301 | 36.37% | 98% | 123135.8 |
| 2000999 | HBV | B | | 514 | 98 | mNGS | 4,026,934 | 52 | 0.001% | 28% | 2.02 |
| - | - | - | *not detected* | - | - | xGen | 2,155,897 | 169 | 0.01% | 15% | 26 |
| - | HDV | 1 | 1150000 | 1675 | 100 | mNGS | 4,026,934 | 22,098 | 0.55% | 99% | 1426.64 |
| - | - | - | - | - | - | xGen | 2,218,134 | 1,279,533 | 57.69% | 99% | 84602.53 |
| 2001063 | HBV | nd | - | 173 | nd | mNGS | 49,274 | 0 | 0% | 0% | 0 |
| - | - | - | *not detected* | - | - | xGen | 31,248 | 3 | 0.01% | 5% | 1 |
| - | HDV | 1 | 3.03x10(7) | 1531 | 100 | mNGS | 49,274 | 3,141 | 6.37% | 91% | 267.68 |
| - | - | - | - | - | - | xGen | 32,590 | 11,334 | 34.78% | 91% | 920.13 |
| 2001073 | HBV | A2 | - | 2038 | 100 | mNGS | 2,371,838 | 127 | 0.01% | 10% | 1 |
| - | - | - | *25.09* | - | - | xGen | 426,656 | 6,000 | 1.41% | 63% | 174 |
| - | HDV | 1 | 1.26x10(6) | 1675 | 100 | mNGS | 2,918,854 | 11,643 | 0.40% | 94% | 790.47 |
| - | - | - | - | - | - | xGen | 464,694 | 203,456 | 43.78% | 100% | 12967.16 |
| 2001149 | HBV | (B/D dual) | - | 2346 | 100 | mNGS | 2,167,544 | 110 | 0.01% | 8% | 1 |
| - | - | - | *29.18* | - | - | xGen | 3,467,598 | 756 | 0.02% | 69% | 20 |
| - | HDV | 1 | - | 1677 | 100 | mNGS | 2,523,132 | 151,554 | 6.01% | 99% | 10179.87 |
| - | - | - | *16.8* | - | - | xGen | 3,775,760 | 1,236,439 | 32.75% | 99% | 84803.89 |
| 2001167 | HBV | nd | - | nd | nd | mNGS | 3,613,434 | 47 | 0.001% | 22% | 2.23 |
| - | - | - | *not detected* | - | - | xGen | 761,172 | 856 | 0.11% | 2% | 568.18 |
| - | HDV | 1 | - | 1677 | 100 | mNGS | 3,613,434 | 30,647 | 0.85% | 96% | 2109.39 |
| - | - | - | *22.2* | - | - | xGen | 761,172 | 340,999 | 44.80% | 98% | 23133.83 |
| 2001178 | HBV | A | - | 530 | 76 | mNGS | 2,601,041 | 515 | 0.02% | 1% | 1 |
| - | - | - | *not detected* | - | - | xGen | 21,182 | 568 | 2.68% | 16% | 75 |
| - | HDV | 1 | - | 1526 | 100 | mNGS | 3,387,460 | 669 | 0.02% | 95% | 45.54 |
| - | - | - | - | - | - | xGen | 23,480 | 12,194 | 51.93% | 91% | 841.31 |
| 2001190 | HBV | D basal | - | 486 | 75 | mNGS | 3,579,444 | 129 | 0.004% | 18% | 7.38 |
| - | - | - | *not detected* | - | - | xGen | 2,662,835 | 38 | 0.001% | 15% | 4 |
| - | HDV | 1 | - | 1683 | 100 | mNGS | 3,579,444 | 130144 | 3.64% | 100% | 8261.17 |
| - | - | - | *17.17* | - | - | xGen | 2,913,414 | 1382215 | 47.44% | 100% | 88874.32 |
| 2001210 | HBV | nd | - | nd | nd | mNGS | 4,217,318 | 56 | 0.001% | 34% | 1.73 |
| - | - | - | *not detected* | - | - | xGen | 99,342 | 0 | 0% | 0% | 0 |
| - | HDV | 1 | - | 1538 | 100 | mNGS | 4,217,318 | 1232 | 0.03% | 92% | 85.4 |
| - | - | - | *24.7* | - | - | xGen | 99,342 | 59937 | 60.33% | 93% | 4347.25 |
| 2001212 | HBV | nd | 39 IU/ml | 239 | nd | mNGS | 1,866,015 | 65 | 0.003% | 7% | 2 |
| - | - | - | - | 295 | 40 | xGen | 851,225 | 681 | 0.08% | 9% | 147 |
| - | HDV | 1 | 5.3x10(6) | 1521 | 100 | mNGS | 2,377,234 | 2694 | 0.11% | 89% | 198.87 |

(*Continued*)

**Table 2.** (Continued)

| Specimen ID | Virus | Genotype | Viral load / Ct | Final sequence length (nt) | Bootstrap | Library Type | Total Reads | HBV or HDV reads | % virus reads | Genome Coverage% | Avg coverage depth |
|---|---|---|---|---|---|---|---|---|---|---|---|
| - | - | - | - | - | - | xGen | 866,568 | 509708 | 58.82% | 92% | 34959.53 |
| 2001222 | HBV | nd | - | 110 | nd | mNGS | 1,610,506 | 16 | 0.001% | 8% | 2.11 |
| - | - | - | not detected | - | - | xGen | 522,338 | 1079 | 0.21% | 3% | 266 |
| - | HDV | 1 | 9.11x10(6) | 1423 | 100 | mNGS | 1,610,506 | 1929 | 0.12% | 89% | 157.42 |
| - | - | - | - | - | - | xGen | 532,072 | 343668 | 64.59% | 90% | 26124.36 |
| 2001228 | HBV | nd | - | 244 | nd | mNGS | 5,186,758 | 56 | 0.001% | 32% | 1.91 |
| - | - | - | not detected | - | - | xGen | 3,975,042 | 4 | 0.0001% | 8% | 0.0127 |
| - | HDV | 1 | 7.52x10(7) | 1683 | 100 | mNGS | 5,186,758 | 73594 | 1.42% | 100% | 4697.18 |
| - | - | - | - | - | - | xGen | 3,975,042 | 1393290 | 35.05% | 100% | 91278.85 |
| 2001234 | HBV | D | - | 494 | 86 | mNGS | 2,821,662 | 20 | 0.001% | 13% | 1.7 |
| - | - | - | not detected | - | - | xGen | 14,849 | 44 | 0.30% | 15% | 9 |
| - | HDV | 1 | 1.66x10(5) | 1509 | 100 | mNGS | 2,821,662 | 230 | 0.01% | 86% | 15.5 |
| - | - | - | - | - | - | xGen | 16,784 | 7154 | 42.62% | 90% | 499.78 |

Summary of results for HBV and HDV sequencing of Israeli specimens.

viruses, but only includes probes tiling a single HBV (NC_003977.2: genotype D) and HDV (NC_001653.2: genotype 1) reference sequence. Genome coverage was comparable when titers were high, however at lower levels, sequence diversity impacted HBV detection with the Pan-viral method, stressing the importance of including probes to multiple genotypes (Table 3). For example, both methods obtained 100% coverage of an HBV B2 with a titer of log 5.92 copies/ml, whereas for the B2 strain (log 3.24 copies/ml) that increased from 8% to 84% with xGen, no coverage (0%) was obtained with the Pan viral method.

## HBV and HDV cohort of patients from Israel

The HBV/HDV mNGS libraries from Israel were revisited with xGen selection. Given the particularly low titers for HBV, most samples (n = 15) only obtained 2–20% genome coverage (60–600 nt/3.2kb), but this was still an increase from zero coverage with mNGS and provided an indication that HBV was indeed present (Table 2). When HBV sequences were originally obtained by mNGS, xGen enrichment significantly (28–73%) improved the overall coverage. As an example, sample 2001149 originally had 8% HBV coverage which was extended to 69% with enrichment (Fig 3A). Interestingly, this individual was dually-infected with genotypes B2 and D1. Incomplete coverage and reliance on a consensus sequence created the appearance of a recombinant, however, overlapping sequences from both strains were detected whereas contiguous reads spanning putative 'recombination breakpoints' were not. (Fig 3A). Phylogenetics from the other patients with >50% of the HBV genome revealed infections with genotypes A1, A2, and D (Table 2).

HDV titers were much higher and thus the abundance of viral reads in mNGS libraries were often sufficient to already provide near full-length sequences (Table 2) [15]. Sample 20001234 was representative of most cases, with HDV-xGen providing significant boosts in sequence depth, but with genome coverage only minimally extended (3–11%) (Fig 3B). We obtained 12 full genomes (>95%) and 14 partial (33–92%) sequences. HDV consensus genomes generated ± enrichment were compared to one another and all agreed with >96% identity unless the mNGS initially yielded partial sequences with low coverage (S3 Table). This indicates the additional rounds of amplification required for xGen did not bias the

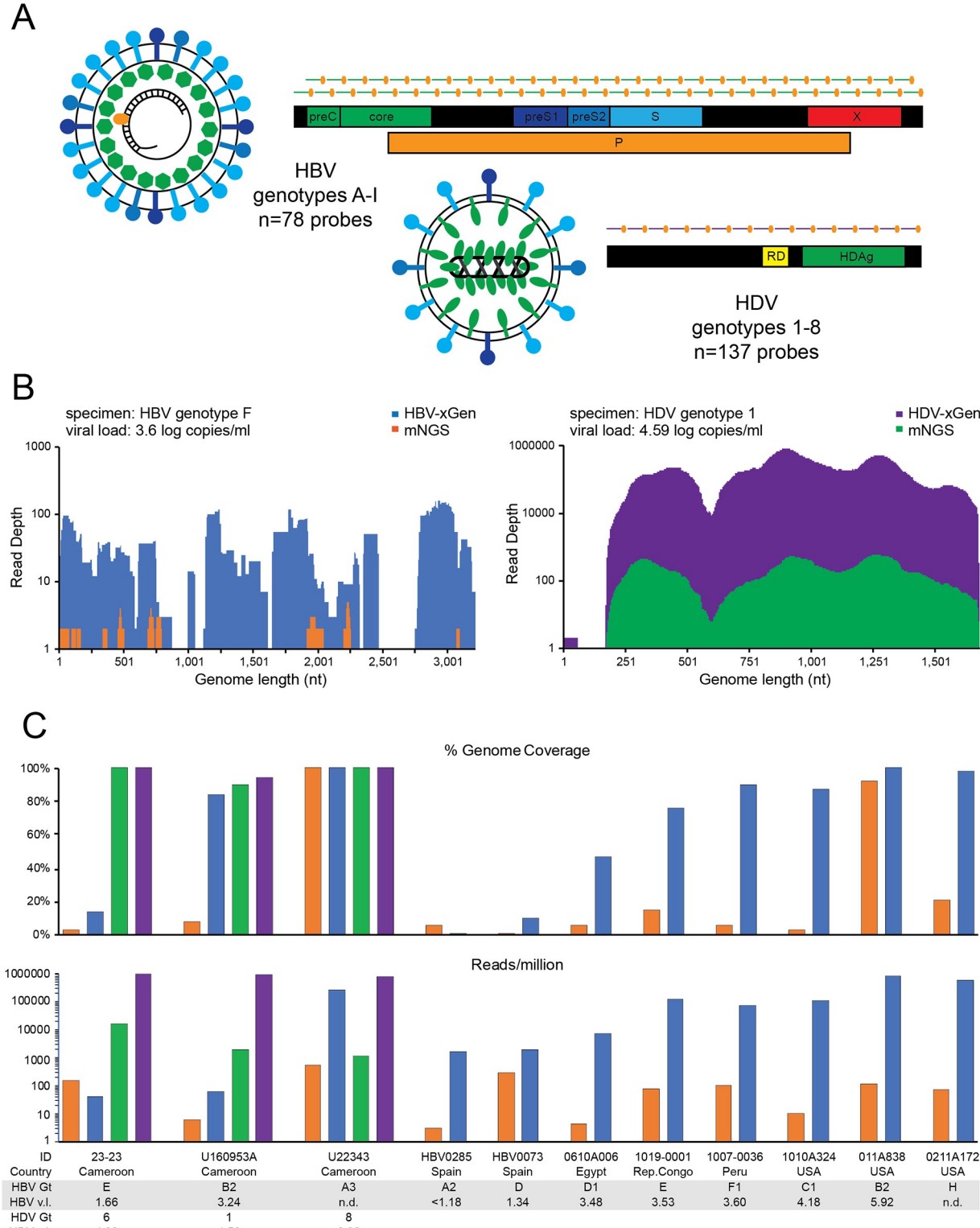

**Fig 2. HBV/HDV-xGen greatly enhances sensitivity.** (A) HBV/HDV xGen probes tile HBV genotypes A-I with 2X coverage and HDV genotypes 1–8 with 1X coverage. (B) Representative genome coverage plots of NGS data for HBV (left) and HDV (right) strains from Cameroon. For HBV, mNGS reads are shown in orange and xGen reads in blue. For HDV, mNGS reads are shown in green and xGen reads in purple. (C) Histograms of coverage (top) and reads/million (bottom) on co-infections or HBV-only infections. Country, genotype, and viral load are listed beneath each plot with the same color scheme as in Fig 2B.

**Table 3. HBV and HDV diversity panel results.**

| Specimen ID | Country | Virus | Genotype | Final length (nt) | Boot strap | VL (log IU/mL) | Library Type | Total Reads | Viral reads | Reads per million | Percent Genome Coverage | Avg coverage depth |
|---|---|---|---|---|---|---|---|---|---|---|---|---|
| 23–23 | Cameroon | HBV | E | 440 | 99 | 1.66 | mNGS | 3,963,160 | 594 | 150 | 3% | 1 |
| - | - | - | - | - | - | - | xGen | 2,284,203 | 95 | 42 | 14% | 16 |
| - | - | HDV | 6 | 1685 | 100 | 4.89 | mNGS | 4,583,420 | 73,543 | 16045 | 100% | 4863 |
| - | - | - | - | - | - | - | xGen | 2,322,206 | 2,270,392 | 977688 | 100% | 137586 |
| U160953A | Cameroon | HBV | B2 | 2731 | 98 | 3.24 | mNGS | 1,624,386 | 10 | 6 | 8% | 1 |
| - | - | - | - | - | - | - | xGen | 3,160,577 | 191 | 60 | 84% | 3 |
| - | - | HDV | 1 | 1554 | 100 | 4.59 | mNGS | 1,624,386 | 3,074 | 1892 | 90% | 212 |
| - | - | - | - | - | - | - | xGen | 3,205,212 | 2,926,340 | 912994 | 94% | 204100 |
| - | - | - | - | - | - | - | Pan viral | 1,162,542 | 390,959 | 336297 | 90% | 369999 |
| U22343 | Cameroon | HBV | A3 | 3221 | 100 | nd | mNGS | 1,583,088 | 859 | 543 | 100% | 19 |
| - | - | - | - | - | - | - | xGen | 3,548,047 | 879,717 | 247944 | 100% | 19624 |
| - | - | HDV | 8 | 1676 | 100 | 6.98 | mNGS | 1,904,026 | 2,136 | 1122 | 100% | 119 |
| - | - | - | - | - | - | - | xGen | 3,628,496 | 2,666,770 | 734952 | 100% | 169534 |
| 0111A838 | USA | HBV | B2 | 3215 | 100 | 5.92 | mNGS | 1,531,143 | 173 | 113 | 92% | 3 |
| - | - | - | - | - | - | - | xGen | 55,902 | 44,229 | 791188 | 100% | 923 |
| - | - | - | - | - | - | - | Pan viral | 690,304 | 40,003 | 57950 | 100% | 1768 |
| 1010A324 | USA | HBV | C1 | 2814 | 100 | 4.18 | mNGS | 1,354,611 | 14 | 10 | 3% | 1 |
| - | - | - | - | - | - | - | xGen | 9,880 | 1,034 | 104656 | 87% | 26 |
| - | - | - | - | - | - | - | Pan viral | 405,484 | 321 | 792 | 78% | 18 |
| 1007-HBV-0036 | Peru | HBV | F1 | 2901 | 100 | 3.6 | mNGS | 2,781,836 | 282 | 101 | 6% | 1 |
| - | - | - | - | - | - | - | xGen | 12,905 | 931 | 72143 | 90% | 27 |
| - | - | - | - | - | - | - | Pan viral | 401,432 | 255 | 635 | 63% | 18 |
| 1019-HBV-0001 | Republic of Congo | HBV | E | 2479 | 100 | 3.53 | mNGS | 1,554,926 | 123 | 79 | 15% | 1 |
| - | - | - | - | - | - | - | xGen | 54,413 | 6,495 | 119365 | 76% | 193 |
| | | | | | | | Pan viral | 22,393,622 | 4,162 | 186 | 70% | 264 |
| 0211A172 | USA | HBV | H | 3178 | 100 | 3.52 | mNGS | 478,506 | 35 | 73 | 21% | 1 |
| - | - | - | - | - | - | - | xGen | 16,763 | 9,313 | 555569 | 98% | 215 |
| - | - | - | - | - | - | - | Pan viral | 147,056 | 1,746 | 11873 | 81% | 96 |
| 0610A006 | Egypt | HBV | D1 | 1496 | 91 | 3.48 | mNGS | 2,582,798 | 11 | 4 | 6% | 2 |
| - | - | - | - | - | - | - | xGen | 56,962 | 406 | 7128 | 47% | 12 |
| - | - | - | - | - | - | - | Pan viral | 424,558 | 291 | 685 | 58% | 22 |
| U160953A | Cameroon | HBV | B2 | 2731 | 98 | 3.24 | mNGS | 1,624,386 | 10 | 6 | 8% | 1 |
| - | - | - | - | - | - | - | xGen | 3,160,577 | 191 | 60 | 84% | 3 |
| - | - | - | - | - | - | - | Pan viral | 1,162,542 | 0 | 0 | 0% | 0 |
| HBV0073 | Spain | HBV | D | 56 | nd | 1.34 | mNGS | 76,111 | 22 | 289 | 1% | 2 |
| - | - | - | - | - | - | - | xGen | 2,044 | 4 | 1957 | 10% | 1 |
| - | - | - | - | - | - | - | Pan viral | 21,630 | 0 | 0 | 0% | 0 |
| HBV0285 | Spain | HBV | A2 | 43 | nd | <1.18 | mNGS | 1,969,686 | 6 | 3 | 6% | 1 |
| - | - | - | - | - | - | - | xGen | 22,230 | 36 | 1619 | 1% | 29 |
| - | - | - | - | - | - | - | Pan viral | 237,712 | 0 | 0 | 0% | 0 |

Summary of results for HBV and HDV sequencing (mNGS, xGen, and Pan viral) for diversity panel specimens.

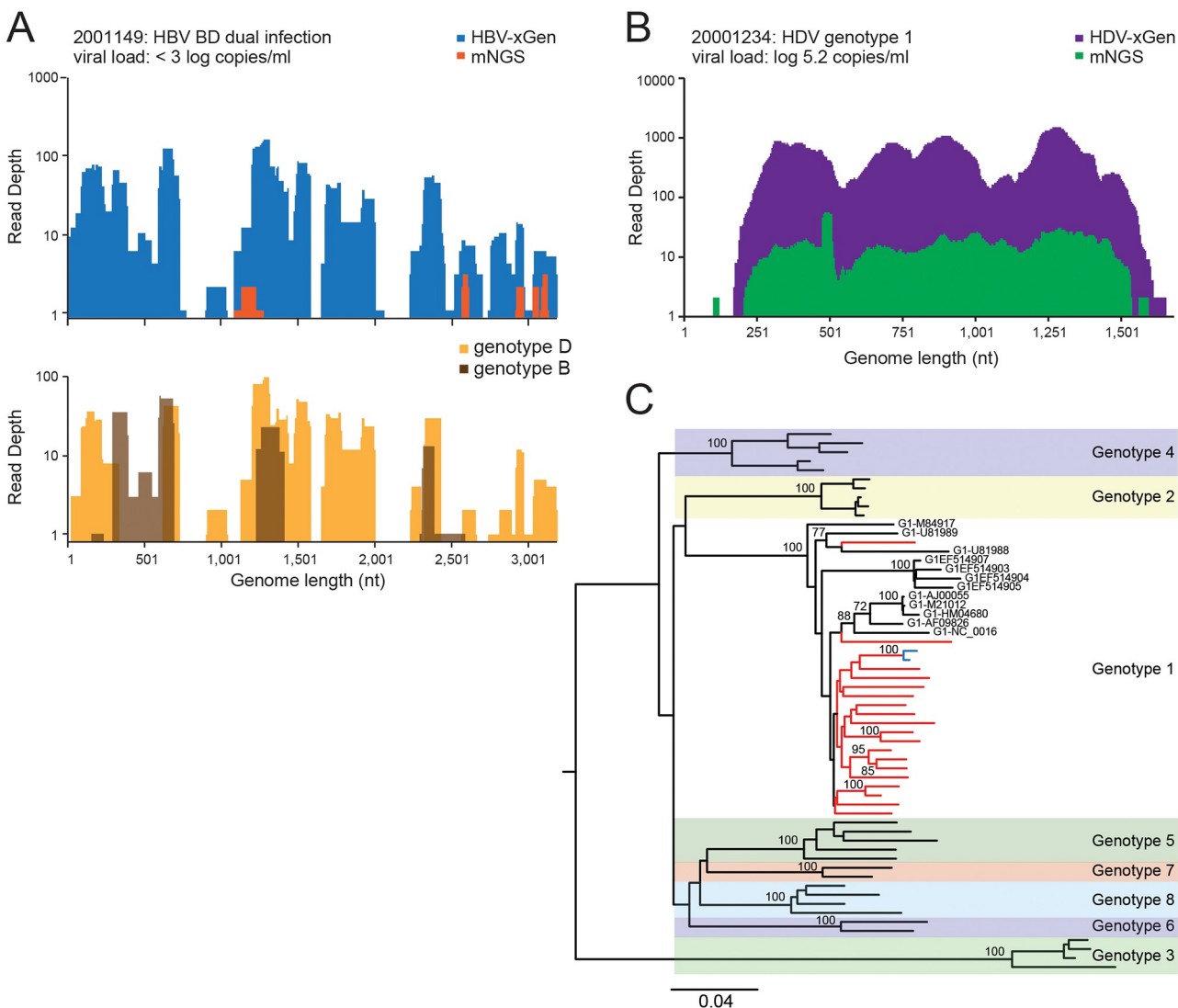

**Fig 3. Dual BD infection and clustering of HDV genotype 1 strains in Israel.** (A) Coverage plot of HBV specimen, 2001149. mNGS reads are shown in orange and xGen reads in dark blue (upper panel). Genotype D1 reads are in gold and genotype B2 reads are in brown (lower panel). (B) Coverage plot of HDV specimen, 2001234. (C) Neighbor-joining phylogenetic tree of near full-length HDV sequences. New Israeli sequence branches are in red and reference strains are in black labeled with accession numbers. Bootstrap values of nodes >70 are shown.

final sequence [13, 28]. Twenty-one strains with >84% genome coverage were included in the phylogenetic analysis. Consistent with previous Sanger sequencing classifications off the HDAg coding region, all strains were HDV genotype 1 (Fig 3C) [15]. Sample 2000742 from Ethiopia branched with HDV strains from Somalia (U81988.1) and Ethiopia (U81989.1). Notably, the HBV sequence from this same individual is genotype A1, indicating both infections likely originated in Africa. Sample 2001222, the only patient from Romania, branches with Gen 1 strains from a variety of geographies, but the rest of the Israeli samples cluster together on a Gen 1 branch absent reference strains from other countries. The 2001149 and 2001063 strains with short branch lengths are from the same patient (blue) and share 99.28% identity.

## Discussion

Metagenomics is an extremely powerful approach that allows one to query for the presence of any pathogen in patient samples without any prior knowledge of the sequence. Yet, due to its unbiased nature, mNGS often only scratches the surface for virus detection, especially for low viral load specimens. Here, the high viral loads typical of HCV, and also with HDV, allowed us to readily obtain full genomes using standard methodology, whereas the low titers of HBV required enrichment [26]. This cohort included 53 males and 43 females with an overall median age of 58. Most were likely exposed to HCV through IDU or blood transfusion, while others had no identifiable risk factor. Sensitivity for numerous viruses in our controls suggested we should have detected other blood borne pathogens if they were there (S1 Fig), but no additional co-infections besides GBV-C were present among the HCV positive individuals.

In Israel, the age-adjusted prevalence of HCV infection was recently estimated to be 5 per thousand, with immigrants from Eastern Europe making up most patients [19]. Indeed, many immigrants to Israel are from countries with high HCV prevalence, such as Georgia, Turkmenistan, Moldova, Uzbekistan, Ukraine, Morocco, Romania, and Kazakhstan. The disparity in HCV infection rates for native-born Israelis (0.1%) versus immigrants (5.7%) is significant and further borne out in studies focusing on IDU populations [18, 29]. While the prevailing trend reported is that HCV Genotype 1b predominates in Israel, the presence of other genotypes besides type 1 (70%) has been noted, including type 2 (8%), type 3 (20%) and type 4 (2%) [16, 19–21, 30]. The majority of HCV strains characterized here were genotype 1b and found in individuals immigrating from the former Soviet Union and neighboring Eastern European countries. Nearly half (10/22) of the native born in Israelis in the cohort were genotype 1a, with the remaining 9 genotype 1a strains coming from Western Europe and the Middle East/North Africa. Genotype 3a strains were primarily from middle aged (49 yr old), IDU males from any country.

Sequence-independent (nuclease treatment, filtration, ultracentrifugation) and -dependent (rRNA depletion, CRISPR cas9-mediated DASH) methods represent enrichment strategies intended to concentrate viral nucleic acid or lower host background [31, 32]. Target enrichment represents an alternate approach based on positive selection which greatly enhances sensitivity for low viral load infections in clinical samples and consequently reduces the overall depth of sequencing required [12, 28, 33]. Comprehensive approaches like ViroCap which tile >185,000 sequences totaling ~200 Mb, include probes against RefSeqs, near-neighbors, and other viral databases, just as VirCapSeq covers 207 viral taxa with nearly 2 million probes, even after clustering highly identical regions [34, 35]. These methods provide amazing boosts in coverage and depth, but these probe sets are prohibitively expensive to synthesize and use on a routine basis. A commercialized, and somewhat leaner probe set (~600,000 probes) covering ~1000 human viruses from Twist Biosciences now puts this technology within reach of more labs. The trade-off we have observed is that while sensitivity for many viruses is substantially improved, performance is compromised for highly diverse viruses where probes are designed against only one or a few representative strains (Table 3). When public health measures require full genome sequencing to halt outbreaks, observe transmission clusters, or track diagnostically relevant mismatches, mere identification of specific viruses like HIV and HCV is insufficient. We and others have shown that probes to all subtypes and groups (HIV) and genotypes (HCV) are required to reliably obtain full genomes [13, 14, 36].

We therefore tailored a specific probe set to ensure capture of all HBV (A-I) and HDV (1–8) genotypes. Boosts in coverage and sequence depth on a variety of HBV specimens with a range of viral loads validated this approach (Fig 2C). xGen did extend coverage for HBV in some co-infected Israeli samples, however at titers this low, there was only so much

improvement to be expected. This suppression of HBV titers by HDV that we observed is consistent with numerous reports [37, 38]. Where HBV sequences were obtained, these were genotype A1, A2, D, and a B/D dual infection. HDV viral loads on the other hand were considerably higher than HBV and enrichment was often not necessary. Our full genome sequencing confirmed previous sub-genomic HDAg protein sequencing wherein genotype 1 was the only strain detected [15]. Levels of co-infection in most countries are simply not known, and as we have seen in Cameroon, they can be far higher than expected [39]. The significant prevalence of HDV (6.5%) in Israel mandates HDV RNA testing for all coinfected patients.

mNGS has yielded a wealth of information and at times, actionable results for patients [40–42]. However, due to costs, turn-around time, validation, and reimbursement among many issues, the arrival of mNGS in the clinic as a test to replace all other infectious disease diagnostics does not appear imminent [43]. Nevertheless, as we demonstrate here, it can play an important role as a research tool and a means of insight into epidemiologic trends. While levels of hepatitis are going down, Israel is home to many immigrants and surveillance is needed. The numerous applications for these and other viruses ensure that mNGS will play an even greater role in dictating public health policy in Israel and elsewhere.

## Supporting information

**S1 Appendix. Pan-viral enrichment protocol.**
(PDF)

**S1 Fig. Viral detection in mNGS libraries.** A positive control consisting of 8 viruses and chlamydia trachomatis spiked into normal human plasma each at log 4.0 copies/ml was included with samples in three separate extractions, library preps and sequencing runs of HCV positive samples. Reads were taxonomically assigned by the SURPI pipeline. The top histogram represents the genome coverages and the bottom histograms represent reads per million for each pathogen.
(PDF)

**S1 Table. Patient demographic data.** Demographic information for HCV and HBV/HDV positive patients enrolled in study at the Israeli National HIV and Viral Hepatitis Reference Center (NHRL).
(PDF)

**S2 Table. HCV mNGS library metrics.** HCV positive library results are listed and include total reads, HCV reads, percent genome coverage, genotype classification, and HCV reads per million.
(PDF)

**S3 Table. HDV sequence agreement.** Pairwise nucleotide identity values comparing HDV consensus sequences from mNGS versus xGen is expressed as a percent.
(PDF)

## Acknowledgments

We thank Guixia Yu, Scot Federman, and Dr. Charles Chiu at University of California San Francisco for sequencing and data processing of HCV samples. We thank Dr. Matthew Frankel for the HBV genotype panel specimens. We thank Dave Campbell and Dr. Nicholas Downey at IDT for assistance with xGen probe design, and John Robichaud and Mark Consugar at Twist Biosciences for assistance with the Pan-viral protocol.

## Author Contributions

**Conceptualization:** Michael G. Berg, Julie Yamaguchi.

**Data curation:** Michael G. Berg, Ana Olivo, Kenn Forberg, Barbara J. Harris, Rachel Shirazi.

**Formal analysis:** Michael G. Berg, Ana Olivo, Barbara J. Harris, Yael Gozlan, Mary A. Rodgers.

**Methodology:** Michael G. Berg, Ana Olivo, Kenn Forberg, Julie Yamaguchi, Rachel Shirazi, Yael Gozlan.

**Project administration:** Michael G. Berg, Orna Mor, Gavin A. Cloherty.

**Resources:** Silvia Sauleda, Lazare Kaptue, Gavin A. Cloherty.

**Supervision:** Michael G. Berg, Mary A. Rodgers, Orna Mor, Gavin A. Cloherty.

**Writing – original draft:** Michael G. Berg.

**Writing – review & editing:** Michael G. Berg, Mary A. Rodgers, Orna Mor, Gavin A. Cloherty.

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
