## [Decision Letter · Decision Letter 0]

1 May 2020

PONE-D-20-06995

Advanced Molecular Surveillance Approaches for Characterization of Blood Borne Hepatitis Viruses

PLOS ONE

Dear Dr. Berg,

Thank you for submitting your manuscript to PLOS ONE. After careful consideration, we feel that it has merit but does not fully meet PLOS ONE’s publication criteria as it currently stands. Therefore, we invite you to submit a revised version of the manuscript that addresses the points raised during the review process.

Your manuscript was reviewed by 2 experts in the field. Although both reviewers found your paper interesting, they identified several issues that require your attention. Please review the attached comments and provide point-by-point responses.

We would appreciate receiving your revised manuscript by Jun 15 2020 11:59PM. To enhance the reproducibility of your results, we recommend that if applicable you deposit your laboratory protocols in protocols.io, where a protocol can be assigned its own identifier (DOI) such that it can be cited independently in the future. For instructions see: http://journals.plos.org/plosone/s/submission-guidelines#loc-laboratory-protocols

We look forward to receiving your revised manuscript.

Kind regards,

Yury E Khudyakov, PhD

Academic Editor

PLOS ONE

Journal Requirements:

2. Please specify whether you obtained IRB approval for use of specimens collected from volunteer blood donors in Spain and Cameroon, and if so, the full name of the IRB that approved this.

3. Please provide additional details regarding participant consent in the ethics statement in the Methods.

Please ensure that you have specified (i) whether consent was informed and (ii) what type you obtained (for instance, written or verbal, and if verbal, how it was documented and witnessed).

If your study included minors, state whether you obtained consent from parents or guardians.

If the need for consent was waived by the ethics committee, please include this information.

4. In your Methods section, please provide additional information about the participant recruitment method and the demographic details of your participants. Please ensure you have provided sufficient details to replicate the analyses such as:

a) the recruitment date range (month and year),

b) a description of any inclusion/exclusion criteria that were applied to participant recruitment,

c) a table of relevant demographic details,

d) a statement as to whether your sample can be considered representative of a larger population, and

e) a description of how participants were recruited.

7. Thank you for stating the following in the Financial Disclosure section:

'The authors were funded by Abbott Laboratories.'

We note that one or more of the authors have an affiliation to the commercial funders of this research study, Abbott Laboratories.

Reviewers' comments:

Reviewer's Responses to Questions

**Comments to the Author**

1. Is the manuscript technically sound, and do the data support the conclusions?

Reviewer #1: Yes

Reviewer #2: Yes

2. Has the statistical analysis been performed appropriately and rigorously? 

Reviewer #1: N/A

Reviewer #2: N/A

3. Have the authors made all data underlying the findings in their manuscript fully available?

Reviewer #1: Yes

Reviewer #2: Yes

4. Is the manuscript presented in an intelligible fashion and written in standard English?

Reviewer #1: Yes

Reviewer #2: Yes

5. Review Comments to the Author

Reviewer #1: The present manuscript reports on HCV, HBV, and HDV molecular epidemiology in infected patients from Israel by using the NGS MiSeq system to obtain full genome sequences. HCV sequencing was successfully performed directly on 99 samples carrying relatively high viral RNA loads as expected. Phylogenetic analysis indicated that genotype 1 was dominant in these patients (clinical status?) with subgenotype differences between patients born in Israel (HCV-1a) and patients originating from Russia and Eastern Europe. Diversity was illustrated further by the presence of genotypes 2-4. It would be interesting to know if there was difference in genome coverage and HCV reads/million according to genotypes. It is unexpected that the authors did not report the detection of any pegivirus which are quite prevalent in humans.

In parallel, 26 samples previously documented HBV/HDV co-infected samples were investigated. Due to the extremely low level of HBV DNA in these samples, two methods of viral targets enrichment were evaluated. xGen probe sets (2x coverage) for the main HBV and HDV genotypes were designed and combined in the same assay. The method was first evaluated on a panel of characterized HBV mono-infected and HBV/HDV dual-infected samples. The results obtained with both panel and patient samples showed the efficiency of the method to recover HBV sequences. The use of a commercial pan viral probe set for HBV was less efficient showing the importance to use probes covering the genetic diversity of all HBV genotypes. Target enrichment showed poor added value for HDV sequencing. The HBV and HDV genotypes observed were consistent with previous data from the region. An interesting feature is the identification of a rare HBV-BD recombinant strain. However, the Simplot profile was derived from a consensus sequence generated by NGS. How the authors excluded the possibility of dual infection? Did they check that there was no overlapping reads characteristic of the two genotypes?

Reviewer #2: In this manuscript, the authors presented a metagenomic next generation sequencing approach (mNGS) for the molecular characterization of blood borne viral hepatitis viruses (HCV, HBV and HDV).

Overall, the methods employed by the authors to accomplish their objectives were adequate and well-presented. The viral enrichment method, crucial for mNGS and a topic of great interest for virological studies, presented solid results when applied in the international HBV panel but not so consistent regarding Israeli samples. Nonetheless, the technique allowed to detect a previously unreported recombinant B/D strain which was a great achievement. The tables presented in the article should be clearer. It lacks proper legend to understand what hyphen means in many lines (not done, not available?).

The employment of new methodologies to provide tools for virological surveillance is relevant for scientific community and this paper surely contributes to a remarkable bottleneck of viral metagenomics which is the low sensibility of detection of low-titer viral DNA/RNA. However, as highlighted by the authors, its use in routine diagnosis does not appear imminent, especially in low income countries.

The quality of the written English is good thorough out the article.

Minor revision

1. Table 1 is not self-explanatory and should be clearer. Only in the text it is possible to understand that the values shown are either absolute numbers or percentages. This must be clearly indicated in the table.

2. The description of the cohort in line 381 repeats the male gender twice. Authors must correct the typing error.

3. A typo in the word “Kazakhstan” (line 391) must be corrected.

6. PLOS authors have the option to publish the peer review history of their article (what does this mean?). If published, this will include your full peer review and any attached files.

Reviewer #1: No

Reviewer #2: No

---

## [Author Response · Author response to Decision Letter 0]

22 Jun 2020

Reviewer #1: The present manuscript reports on HCV, HBV, and HDV molecular epidemiology in infected patients from Israel by using the NGS MiSeq system to obtain full genome sequences. HCV sequencing was successfully performed directly on 99 samples carrying relatively high viral RNA loads as expected. Phylogenetic analysis indicated that genotype 1 was dominant in these patients (clinical status?) with subgenotype differences between patients born in Israel (HCV-1a) and patients originating from Russia and Eastern Europe. Diversity was illustrated further by the presence of genotypes 2-4. It would be interesting to know if there was difference in genome coverage and HCV reads/million according to genotypes.

The answer can be found in Supplemental Table S1. All the 2c and 3a sequences had >98% genome coverage and >1200 rpm. A few had <97% genome coverage, and these were genotype 1A or 1B, which made up the majority of samples.

It is unexpected that the authors did not report the detection of any pegivirus which are quite prevalent in humans.

We regret the confusion here; an initial version of the text attempted to make the following distinction, which we ought to have kept in. HPgV-2 is a novel pegivirus that our lab discovered in 2015: it is primarily seen in HCV infections, so we were looking specifically for this virus, but we did not detect it in these HCV+ individuals. We did however observe human pegivirus-1, also known as GBV-C, in 5 individuals. As the reviewer notes, it is quite common. We have included the following sentence on lines 264-265, “Human pegivirus-1 (GBV-C) reads were enriched in 5 individuals.” We also further clarify this in the Discussion on lines 389-91,

“Sensitivity for numerous viruses in our controls suggested we should have detected other blood borne pathogens if they were there (S1 Fig), but no additional co-infections besides GBV-C were present among the HCV positive individuals.”

In parallel, 26 samples previously documented HBV/HDV co-infected samples were investigated. Due to the extremely low level of HBV DNA in these samples, two methods of viral targets enrichment were evaluated. xGen probe sets (2x coverage) for the main HBV and HDV genotypes were designed and combined in the same assay. The method was first evaluated on a panel of characterized HBV mono-infected and HBV/HDV dual-infected samples. The results obtained with both panel and patient samples showed the efficiency of the method to recover HBV sequences. The use of a commercial pan viral probe set for HBV was less efficient showing the importance to use probes covering the genetic diversity of all HBV genotypes. Target enrichment showed poor added value for HDV sequencing. The HBV and HDV genotypes observed were consistent with previous data from the region. An interesting feature is the identification of a rare HBV-BD recombinant strain. However, the Simplot profile was derived from a consensus sequence generated by NGS. How the authors excluded the possibility of dual infection? Did they check that there was no overlapping reads characteristic of the two genotypes?

We thank the reviewer for their very astute inquiry. After careful scrutiny of the data, we have reached the conclusion that indeed it is not a recombinant, but rather a dual infection. As they noted, there were overlapping reads indicative of both genotypes, but because a consensus sequence was used for phylogenetics and SIMPLOT analysis, these less abundant reads were obscured by the majority sequence. As the plot in Fig 3A indicated and we noted in the text, coverage was incomplete at 69%. Upon closer inspections, the transition from one genotype to the other occurred at these gaps. In addition, there were no contiguous reads that spanned the putative ‘breakpoints’. Rather, one could see where reads from one genotype ended and a new set of reads for the other genotype began. We regret this oversight and are grateful to the reviewer for raising this possibility and allowing us to catch this mistake.

We have modified the abstract (lines 51-54) to read as follows:

When HBV-xGen was applied to Israeli samples, coverage was improved by 28-73% in 4 samples and identified HBV genotype A1, A2, D1 specimens and a previously never described BD recombinant dual B/D infection.

We have modified the results (lines 339-343):

Before

Interestingly, this strain was classified as a rare BD recombinant, which we confirmed by bootscanning in SIMPLOT and individual trees of fragments (Fig 3B; S2 Fig). A separate bleed from this individual did not yield HBV sequence, but prior PCR confirmed D1 sequence in the RT region.

After

Interestingly, this individual was dually-infected with genotypes B2 and D1. Incomplete coverage and reliance on a consensus sequence created the appearance of a recombinant, however, overlapping sequences from both strains were detected whereas contiguous reads spanning putative ‘recombination breakpoints’ were not. (Fig 3A). 

Note that the original Fig 3B has been removed, as well the accompanying phylogenetic trees of sub-fragments in Fig S2. In its place, Fig 3A now includes a lower panel that illustrates which HBV genotype reads are coming from.

The legend for Fig 3A now reads as follows:

 (A) Coverage plot of HBV specimen, 2001149. mNGS reads are shown in orange and xGen reads in dark blue (upper panel). Genotype D1 reads are in gold and genotype B2 reads are in brown (lower panel).

We have modified the Discussion now to remove two sentences (lines 437-441) and it reads as follows:

Where HBV sequences were obtained, these were genotype A1, A2, D, and a BD recombinant B/D dual infection. Recombinants of HBV are themselves rare, and a BD mosaic genome has never been described until here (Fig 3B, S2 Fig) [39]. Consistent with reports that recombination sites are not distributed randomly in HBV, we observed breakpoints within in the 1700-2000 region. 

Reviewer #2: In this manuscript, the authors presented a metagenomic next generation sequencing approach (mNGS) for the molecular characterization of blood borne viral hepatitis viruses (HCV, HBV and HDV).

Overall, the methods employed by the authors to accomplish their objectives were adequate and well-presented. The viral enrichment method, crucial for mNGS and a topic of great interest for virological studies, presented solid results when applied in the international HBV panel but not so consistent regarding Israeli samples. Nonetheless, the technique allowed to detect a previously unreported recombinant B/D strain which was a great achievement. The tables presented in the article should be clearer. It lacks proper legend to understand what hyphen means in many lines (not done, not available?).

The employment of new methodologies to provide tools for virological surveillance is relevant for scientific community and this paper surely contributes to a remarkable bottleneck of viral metagenomics which is the low sensibility of detection of low-titer viral DNA/RNA. However, as highlighted by the authors, its use in routine diagnosis does not appear imminent, especially in low income countries.

The quality of the written English is good thorough out the article.

Minor revision

1. Table 1 is not self-explanatory and should be clearer. Only in the text it is possible to understand that the values shown are either absolute numbers or percentages. This must be clearly indicated in the table.

The columns headers in Table 1 have been modified to indicate the absolute number of total individuals.

2. The description of the cohort in line 381 repeats the male gender twice. Authors must correct the typing error.

It now reads “…53 males and 43 females…”

3. A typo in the word “Kazakhstan” (line 391) must be corrected.

Corrected.

---

## [Editor Report · Decision Letter 1]

29 Jun 2020

Advanced Molecular Surveillance Approaches for Characterization of Blood Borne Hepatitis Viruses

PONE-D-20-06995R1

Dear Dr. Berg,

We’re pleased to inform you that your manuscript has been judged scientifically suitable for publication and will be formally accepted for publication once it meets all outstanding technical requirements.

Kind regards,

Yury E Khudyakov, PhD

Academic Editor

PLOS ONE
---

## [Editor Report · Acceptance letter]

6 Jul 2020

PONE-D-20-06995R1 

Advanced Molecular Surveillance Approaches for Characterization of Blood Borne Hepatitis Viruses 

Dear Dr. Berg:

I'm pleased to inform you that your manuscript has been deemed suitable for publication in PLOS ONE. Congratulations! Your manuscript is now with our production department. 

Kind regards, 

on behalf of

Dr. Yury E Khudyakov 

Academic Editor

PLOS ONE